# Synergistic Antioxidant and Anti-Inflammatory Activities of Kale Juice Fermented with *Limosilactobacills reuteri* EFEL6901 or *Limosilactobacills fermentum* EFEL6800

**DOI:** 10.3390/antiox12101850

**Published:** 2023-10-12

**Authors:** Ga Yun Kim, Seul-Ah Kim, Sun Young Kong, Hyunbin Seong, Jae-Han Bae, Nam Soo Han

**Affiliations:** Brain Korea 21 Center for Bio-Health Industry, Division of Animal, Horticultural, and Food Science, Chungbuk National University, Cheongju 28644, Republic of Korea; gayun.kim@pulmuone.com (G.Y.K.); sakim@chungbuk.ac.kr (S.-A.K.); ksyjangmi@gmail.com (S.Y.K.); adsm06@cbnu.ac.kr (H.S.); wassup7@hanmail.net (J.-H.B.)

**Keywords:** kale juice, probiotics, *Limosilactobacillus reuteri*, *Limosilactobacillus fermentum*, anti-inflammatory

## Abstract

This study investigates the synergistic impact of fermenting kale juice with *Limosilactobacillus* strains on its antioxidant and anti-inflammatory properties. Kale’s rich nutrient profile, especially its flavonoids, offers potential health benefits. Probiotic lactic acid bacteria are employed in kale fermentation to enhance nutrient bioavailability and generate bioactive compounds. Kale juices fermented with *L. reuteri* EFEL6901 or *L. fermentum* EFEL6800 exhibited superior microbial growth. Free sugars and amino acids were converted to alcohols and organic acids, affecting the organoleptic and health-related properties of the product. In addition, fermentation increased quercetin and kaempferol content, indicating improved availability. Furthermore, the fermented juice exhibited notable antioxidant activity and suppressed nitric oxide (NO) production, revealing anti-inflammatory potential. Gene expression analysis confirmed reduced pro-inflammatory markers such as iNOS, COX-2, IL-6, and IL-1β and elevated anti-inflammatory cytokines, including IL-10. This research highlights the promising potential of fermented kale juice, enriched with *Limosilactobacillus* strains, as a functional food with combined antioxidant and anti-inflammatory benefits.

## 1. Introduction

The development of novel nondairy-fermented products utilizing vegetables, fruits, legumes, or cereals has garnered significant attention in recent times [1]. These products not only exhibit probiotic properties but also serve as abundant sources of phytochemicals and phytonutrients [2,3]. Simultaneously, there has been a growing interest in fermented juices made from fruits or vegetables among health-conscious consumers and researchers, driven by the increasing popularity of veganism [4]. Fermented vegetable juices have emerged as a promising category within the functional foods market, offering not only nutritional benefits but also the potential for innovative flavors [5]. This trend aligns with the rising consumer demand for healthier and more sustainable food choices while also addressing the need for convenient and versatile products suited to today’s fast-paced lifestyle [6].

Kale, a vegetable belonging to the genus *Brassica*, species *Brassica oleracea*, group *acephala*, appears frequently in the ‘healthiest foods’ or ‘superfoods’ [7]. This leafy green vegetable is highly regarded for its abundant nutritional value, which encompasses flavonoids, antioxidant enzymes, and other trace compounds [8]. Notably, kale stands out for its rich content of flavonoids, specifically quercetin and kaempferol, which surpass the levels found in other vegetables such as collard greens, mustard greens, sweet potato greens, and green onions [9]. These components exhibit significant potential, not only for disease prevention but also as therapeutic agents. Previous studies suggested that kale may contribute significantly to the management of chronic conditions such as obesity, cholesterol reduction, and inflammatory bowel disease [10].

The concept of a starter culture involves the addition of one or more microorganisms to raw materials, facilitating and controlling the fermentation process in the production of fermented foods [11]. Particularly, lactic acid bacteria (LAB) are widely employed as starter cultures for vegetable fermentations, such as kimchi, sauerkraut, and tempeh [12]. LAB can enhance the bioavailability and digestibility of nutrients present in vegetables, making them more easily absorbed by the body, and producing bioactive metabolites that exhibit antioxidant properties [13,14]. Moreover, LAB can contribute to sensory properties by breaking down various food components in vegetables into flavor precursors and converting them into aromatic compounds, including acetoin and diacetyl [15].

Inflammation is a crucial process that safeguards the host against external threats, such as bacteria, viruses, and toxins, by eliminating pathogens and promoting tissue recovery [16]. However, prolonged inflammation can disrupt immune tolerance, cause significant physiological changes in tissues, organs, and normal cells, and increase the risk of various non-communicable diseases [17]. Kale contains phytochemicals, sulfur-containing indolic glucosinolates, and aliphatic glucosinolates that have demonstrated anti-inflammatory activity [18]. Phenolic compounds such as kaempferol and quercetin have been shown to possess anti-inflammatory effects by inhibiting the expression of PGE2, COX-2, and mPGES-1, which are upregulated by lipopolysaccharide (LPS)-induced inflammation [19]. Moreover, probiotics are widely used for the management of chronic diseases due to their ability to modulate the immune system and elicit anti-inflammatory responses [20,21]. Probiotic strains have shown anti-inflammatory effects by reducing the expression of pro-inflammatory cytokines, primarily through toll-like receptor (TLR)-mediated mechanisms [22]. Previous studies have reported that fermentation of kale juice using *L. acidophilus* IFO 3025 led to improvements in nutrient content, including calcium, phosphorus, and magnesium [6]. Fermentation of kale juice by various LABs has also been found to enhance its antioxidant activity and exhibit antibacterial effects, particularly against Gram-negative bacteria, during the fermentation process, thereby enhancing its physiological properties [23].

This study aimed to develop an enhanced antioxidative and anti-inflammatory product by fermenting kale juice with selected probiotic strains and assessing their health-promoting effects. Specifically, we utilized *Limosilactobacillus reuteri* EFEL6901 [24], *L. fermentum* EFEL6800 [24], *L. fermentum* MG7011 [25], and *Leuconostoc citreum* EFEL2061 [26], which have previously demonstrated probiotic properties with antioxidant and anti-inflammatory activities in relevant research. The fermentation characteristics of these strains were analyzed in kale juice, and the metabolites and flavonoid contents, such as kaempferol and quercetin, were measured. Moreover, the health functionality of the fermented kale juice was evaluated, including its anti-inflammatory and antioxidant activities.

## 2. Materials and Methods

### 2.1. Bacterial Strains and Growth Conditions

The bacterial strains used in this study are presented in Table 1. All strains were preserved in a deep freezer at −80 °C in 50% glycerol (30% *v*/*v*). The strains were cultured in MRS broth (BD Difco, Franklin Lakes, NJ, USA) for 24 h under optimal anaerobic conditions. To initiate the juice fermentation process, the cultured strains were harvested through centrifugation (10,000× *g* for 1 min), washed twice with sterile 0.85% (*w*/*v*) NaCl, and used as inoculums.

### 2.2. Characteristics of Fermented Kale Juices

Kale (*Brassica oleraceae* L. var. *acephala* DC.) was obtained from a local organic market in Cheongju, Korea. The leaves were trimmed to remove damaged portions and washed with tap water. A total of 600 g of kale was then ground using a grinder and mixed with 1800 mL of distilled water. The mixture was pasteurized at 70 °C for 30 min and inoculated with a 4% starter culture consisting of *L. reuteri* EFEL6901, *L. fermentum* EFEL6800, *L. fermentum* MG7011, and *Le. citreum* EFEL2061 at a final concentration of 10^6^ CFU/mL. The inoculated samples were incubated at 37 °C for 72 h. Viable cell counts were determined by serial dilution with 0.85% NaCl and plating on MRS agar medium. The plates were incubated at 37 °C for 48 h, and the viable cell counts were expressed as a colony-forming unit (CFU)/mL. The pH of the kale juices was measured using a pH meter (Orion Star A211, Thermo Fisher Scientific Inc., Waltham, MA, USA), and the titratable acidity was determined by titrating with 0.01 N NaOH, expressed as the percentage of lactic acid in the samples. Color plays a crucial role in food acceptance, and preserving the distinct color after fermentation is important. For this, color measurements of kale juices were performed using a colorimeter (Minolta CR-300, Minolta, Osaka, Japan) with calibration using white and black standard tiles. The samples were placed in transparent cuvettes for color detection, and the chromatic values L*, a*, and b* were recorded. Higher values of L*, a*, and b* indicate higher brightness, redness, and yellowness, respectively. The total color difference (ΔE*) of the samples was calculated using the following formula: ${∆E}^{*}={{(∆L}^{*2}+{∆a}^{*2}+{∆b}^{*2})}^{1/2}\;$. All measurements and analyses were performed in triplicate to ensure accuracy and reproducibility.

### 2.3. DPPH Radical Scavenging Activity Assay

Fermented kale juices were centrifuged at 11,000× *g* for 10 min at 4 °C, and the supernatants were collected. The DPPH (2,2-diphenyl-1-picrylhydrazyl) radical scavenging activities were measured according to the method of Hur [27]. A 200 μL volume of 0.2 mM DPPH solution (in methanol) was added to 50 μL of the fermented kale juice sample, followed by incubation in the dark for 30 min. The absorbance was measured at 517 nm. The various concentrations of ascorbic acid (1 mM, 0.5 mM, and 0.25 mM) were used as positive controls. The DPPH radical scavenging activities were calculated according to the following equation:$$\mathrm{DPPH}\;\mathrm{scavenging}\;\mathrm{activity}\;(\%)=(1-(\frac{{A\;sample}_{}}{{A\;control}_{}}))\times 100$$
where *A sample* is the absorbance of the sample added group; *A control* is the absorbance of the distilled water added group.

### 2.4. ABTS Radical Scavenging Activity Assay

The ABTS+ scavenging activities of samples were measured by modifying the method of Biglar [28]. ABTS+ radical cations were generated by reacting 7 mM 2,2′-azobis (2-amidinopropane) dihydrochloride and 2.45 mM potassium persulfate in distilled water at room temperature for 24 h in the dark. Before analysis, the ABTS+ solution was diluted with distilled water to adjust an absorbance of 1.00 ± 0.02 at 734 nm. Consecutively, 10 μL of fermented kale juices and 200 μL of ABTS+ solution were mixed, and the absorbances were measured at 734 nm after the reaction for 6 min. The various concentrations of ascorbic acid (1 mM, 0.5 mM, and 0.25 mM) were used as positive controls. The ABTS+ radical scavenging activities were calculated according to the following equation:$$\mathrm{ABTS}\;\mathrm{radical}\;\mathrm{scavenging}\;\mathrm{activity}\;(\%)=(1-(\frac{{A\;sample}_{}}{{A\;control}_{}}))\times 100\;$$
where *A sample* is the absorbance of the sample added group; *A control* is the absorbance of the distilled water added group.

### 2.5. Quercetin and Kaempferol Analysis

To analyze the quercetin and kaempferol, the lyophilized fermented kale juice sample was extracted with 50% aqueous methanol and 1.5 M HCl (37.5 mL of 66.7% aqueous methanol and 12.5 mL of 6 M HCl) for twice. For freeze-drying, samples were frozen at −80 °C for 12 h and lyophilized by using a freeze dryer FDU-1200 (EYELA, Tokyo, Japan). Furthermore, it was refluxed at 90 °C for 2 h, and the extract was filtered through a 0.45 μm PTFE filter for HPLC analysis [29]. HPLC analysis of the flavonoids was conducted using an Infinity 1260 HPLC (Agilent Technologies, Santa Clara, CA, USA) consisting of an autosampler, the Phoroshell C18 column (4.6 × 150 mm, 120 Å, 2.7 µm) (Agilent technologies), and a UV detector. The measurement was performed using the method of Rha [30]. In detail, the flow rate was set at 0.8 mL/min, and the gradient elution was made with mobile phase A (0.1% formic acid in water) and mobile phase B (0.1% formic acid in acetonitrile). The elution gradient was as follows: 0–8% B (0–2 min), 8–12% B (2–3 min), 12–16% B (3–4 min), 16% B (4–12 min), 16–20% B (12–15 min), 20% B (15–18 min), 20–24% B (18–21 min). 24–30% B (21–22 min), 30% B (22–26 min), 30–50% B (26–28 min), 50% B (28–30 min), 50–80% B (30–32 min), 80% B (32–33 min), 80–8% B (33–34 min), and 8% B (34–35 min). The UV chromatograms recorded at 365 nm were used for the quantification of quercetin (≥95%, Sigma-Aldrich, St. Louis, MO, USA) and kaempferol (≥90%, Sigma-Aldrich), based on external standard calibration curves. The standard curve for quantification was prepared by serially diluting the standard substances quercetin and kaempferol from 100 to 12.5 μM.

### 2.6. ^1^H-NMR Analysis

To analyze the metabolites of fermented kale juices, a nuclear magnetic resonance (NMR) spectroscopy analysis was conducted. For the preparation of samples, fermented kale juices were centrifuged (11,000× *g*, 10 min), and then the supernatants were collected. The pH of the supernatants was adjusted to 6.0 ± 0.2 and mixed with 350 μL of 1 mM sodium 2,2-dimethyl-2-silapentane-5-sulfonate (DSS; Sigma-Aldrich) as an internal standard in 10% deuterium oxide (D_2_O; Sigma-Aldrich). The mixtures (700 μL) were transferred to NMR tubes and subjected to ^1^H-NMR analysis. The ^1^H-NMR spectra were recorded on an Avance 500-MHz spectrometer (Bruker BioSpin, Karlsruhe, Germany). The metabolite peaks in the NMR spectra were identified, and their concentrations were calculated using Chenomx NMR suite 8.4 library software (Chenomx Inc., Edmonton, AB, Canada).

### 2.7. Cytotoxicity of Fermented Kale Juice

The cytotoxicity was assessed by the MTT (3-(4,5-dimethylthiazol-2-yl)-2,5-diphenyltetrazolium bromide) assay on RAW 264.7 cell lines. The RAW 264.7 cell lines were obtained from the Korean Cell Line Bank (Seoul, Korea). Briefly, RAW 264.7 cells (2.0 × 10^5^ cells/well) were seeded in 96-well plates with DMEM containing 10% FBS and cultured at 37 °C for 18–20 h. Furthermore, 6.25–100 μg/mL of lyophilized kale juice powder dissolved in DMSO and 1 μg/mL of LPS (Sigma-Aldrich) were treated with the cells simultaneously for 24 h. After incubation, the medium was removed, and cells were washed twice with PBS. Furthermore, 100 μL of MTT solution (0.25 mg/mL) (HiMedia, Mumbai, India) was added, and the cells were additionally incubated in the dark at 37 °C for 1 h. After cultivation, the supernatant containing the MTT reagent was carefully removed, and the formazan crystals were dissolved by adding 150 μL of DMSO. Finally, the absorbance of each well was measured at 570 nm using a microplate spectrophotometer (BioTek, Santa Barbara, CA, USA). The percent cell viability was calculated according to the following equation: $$\mathrm{Cell}\;\mathrm{viability}\;(\%)=\{1-(\frac{absorbance\;in\;the\;sample\;treated\;with\;1\frac{\mu g}{mL}LPS}{absorbance\;in\;the\;treated\;1\frac{\mu g}{mL}LPS})\}\times 100$$

### 2.8. Measurement of Nitric Oxide (NO) Production

The effect of kale juices fermented with different starters on the production of NO was determined in LPS-induced RAW 264.7 cell lines using the Griess reaction [31]. RAW 264.7 cells (2 × 10^5^ cells/well) were seeded in 96-well plates and incubated for 18–20 h. RAW 264.7 cells were treated with fermented kale juice at the above-mentioned concentrations, followed by stimulation with LPS (1 μg/mL). Methyl arginine was used as the positive control. After 24 h of incubation, 100 μL of culture supernatants were transferred to 96-well plates, mixed with an equal volume of Griess reagent (Sigma-Aldrich), and placed in the dark for 15 min at room temperature. The absorbance of each mixture at 540 nm was measured using a microplate spectrophotometer (BioTek). Nitrite concentration was calculated using dilutions of sodium nitrite as standards, and fresh medium was used as the blank control.

### 2.9. mRNA Expression Level of Nitric Oxide Synthase (iNOS), Cyclooxygenase-2 (COX-2), and Cytokine

The effect of fermented kale juice on the transcription levels of pro-inflammatory and anti-inflammatory genes was evaluated in LPS-induced RAW 264.7 cells using RT-qPCR. RAW 264.7 cells (1 × 10^6^ cells/well) were seeded in 6-well plates and incubated for 24 h. Freeze-dried kale juice powder inoculated with different strains (in DMSO) and 1 μg/mL of LPS were simultaneously treated with the cells for 24 h. After incubation, cells were washed twice with sterile PBS, and total RNA was extracted with the Trizol RNA isolation reagent (Invitrogen, Carlsbad, CA, USA) according to the manufacturer’s protocol. A total of 200 ng of RNA from each sample was utilized for reverse transcription to cDNA using the LeGene Express 1st Strand cDNA Synthesis System Kit (LeGene Biosciences, San Diego, CA, USA). The Ct value of the GAPDH gene was employed to verify the uniformity of the reverse transcription process across different samples. Real-time PCR was conducted using the Exicycler 96 Real-Time Quantitative Thermal Block (Exicycler 96; Bioneer, Daejeon, Korea). The mixtures containing synthesized cDNA, 10 pmol of specific primers, and PCR Master Mix containing SYBR Green Mix were amplified as follows: 95 °C for 5 min, followed by 45 cycles at 95 °C for 15 s, 59 °C for 30 s, and a final extension at 59 °C for 30 s. The result was analyzed after normalization with GAPDH as a reference gene. Relative expression levels of target genes were calculated with the delta-delta Ct (ΔΔCt) method. The specific primer sequences used in this study are listed in Table 2.

### 2.10. Enzyme and Related Genes

Genetic analysis was conducted to identify the genes related to the utilization of phytochemicals through a sequence similarity search. For this, the whole genome sequences of *L. reuteri* EFEL6901 (Accession number: CP095735.1) and *L. fermentum* EFEL6800 (Accession number: CP124737.1) were obtained from the NCBI database (https://www.ncbi.nlm.nih.gov/, accessed on 4 October 2023). Furthermore, genes related to glycoside hydrolase or glucosidase enzymes were identified using BLASTP. The detailed mechanisms of enzymes were predicted by CAZypedia, available at https://www.cazypedia.org/index.php (accessed on 4 October 2023).

### 2.11. Statistical Analysis

Each experiment was conducted in triplicate, and the data were presented as the mean value ± standard deviation (SD). Statistical analysis was performed using IBM SPSS software version 22 (SPSS Inc., Chicago, IL, USA). Independent *t*-tests were employed to analyze the differences between the two groups. Descriptive statistics (mean and standard deviation) were conducted to assess differences among multiple groups. Prior to conducting a one-way ANOVA, the Shapiro–Wilk test was used to assess the normality of kale juice metabolite data. A one-way analysis of variance (ANOVA) followed by Tukey’s test (for normally distributed data) was employed to compare significant differences in metabolite changes between the before and after fermentation with different starter strains. All *p*-values were corrected using the FDR correction method [32].

## 3. Results 

### 3.1. Quality Characteristics of Fermented Kale Juice 

#### 3.1.1. Microbial Growth in Kale Juice 

To evaluate the growth characteristics of different bacterial strains in fermented kale juice, four experimental groups were prepared: kale juice fermented with *L. reuteri* EFEL6901 (KJ-EFEL6901), *L. fermentum* EFEL6800 (KJ-EFEL6800), *L. fermentum* MG7011 (KJ-MG7011), and *L. citreum* EFEL2061 (KJ-EFEL2061). In the preliminary experiment, no bacteria were detected when the samples were pasteurized at 70 °C for 30 min (d). Four strains were inoculated at a concentration of 10^6^ CFU/mL in pasteurized kale juice, and their growth rates were monitored under anaerobic conditions at 37 °C for 72 h. As shown in Figure 1A, all strains exhibited robust growth in kale juice. Notably, EFEL6901 and EFEL6800 demonstrated the most significant increase in cell count, reaching up to 9.0 Log CFU/mL, which corresponded to a 200-fold increment. The pH of the kale juice decreased progressively as the fermentation time advanced, as illustrated in Figure 1B. After 72 h of fermentation, the pH levels ranged from 4.5 to 4.8, depending on the strain. The total acidity of the kale juice also increased during fermentation, surpassing 0.10% in the case of EFEL6901, as shown in Figure 1C. These findings indicate that four bacterial strains are capable of propagating in kale juice, particularly EFEL6901 and EFEL6800, which exhibit more adapted characteristics in kale.

#### 3.1.2. Chromaticity Analysis

To investigate the color changes during the fermentation, kale juice samples were analyzed using a colorimeter, and the results were indicated as lightness (L*), redness (a*), yellowness (b*), and total color difference (ΔE*) (Table 3). After the pasteurization process, the L*, a*, and b* values exhibited changes, showing an increase in lightness and yellowness. After the fermentation process, those values exhibited variable changes depending on the starters. The L* value remained relatively constant (34.61), except for the KJ-EFEL6901 sample (34.02), indicating minimal variation in lightness. The redness (a*) value showed a significant increase after fermentation. In contrast, there were no significant changes observed in yellowness (b*). The color differences (ΔE*) between pre- and post-fermentation samples demonstrated significant variations, ranging from 5.67 to 6.80. These results indicate that both pasteurization and fermentation can influence the color of kale juice, with the extent of change dependent on the specific strain used.

#### 3.1.3. Quantification of Quercetin and Kaempferol

To investigate the change in the flavonoid contents after kale juice fermentation, samples were extracted by reflux and analyzed by HPLC (Figure 2). As a result, the initial concentrations of quercetin and kaempferol in kale juice were measured at 7.87 μM (45.03 mg/100 g) and 54.46 μM (311.78 mg/100 g), respectively, and these levels remained stable even after pasteurization. Following 72 h of fermentation, the flavonoid contents in four samples showed an increase. In quercetin content, KJ-EFEL6800 exhibited a significant increase, reaching 40.06 μM, which was five-fold higher than the pre-fermentation level. KJ-EFEL6901 showed a three-fold increase in quercetin level (20.82 μM), while KJ-MG7011 and KJ-EFLE206 had slight increases of 11.42 and 11.64 μM, respectively. In kaempferol content, KJ-EFEL6901 and KJ-EFEL6800 exhibited about two-fold increases, reaching 97.82 and 101.59 μM, respectively, compared with the pre-fermentation level. KJ-EFEL2061 also demonstrated increased kaempferol content (78.62 μM). Overall, the flavonoid levels in kale juice remained stable after pasteurization, and their contents were significantly enhanced by the fermentation process, particularly in samples inoculated with EFEL6901 and EFEL6800.

#### 3.1.4. H-NMR Analysis

^1^H-NMR analysis was conducted to analyze the metabolite changes such as carbohydrates, alcohols, organic acids, and amino acids in fermented kale juice. As shown in Figure 3, during the fermentation, carbohydrates and amino acids were consumed as microbial cells grew, whereas alcohols and organic acids were produced with the microbial reduction reaction of nutrients after 72 h fermentation. The detailed contents are presented in Table 4. First, the carbohydrate and amino acid contents in the pasteurized kale juice demonstrated a rich and balanced nutrient composition in kale, which is essential as a microbial growth medium. This result is consistent with the robust growth of LAB observed in this study (Figure 1). In the case of carbohydrates, glucose and fructose were identified as the major free sugars, and their concentrations decreased after fermentation in all kale juice samples. This result indicates that all strains can consume glucose and fructose as major carbon sources in kale juice. Interestingly, the reduction in glucose content was consistent across all strains, while the decrease in fructose varied, suggesting differences in fructose metabolic capability among the strains. In addition, mannitol, ethanol, acetate, and lactate were produced in all samples after fermentation, indicating hetero-lactic metabolic characteristics of the LAB tested in this study. Glucose was mainly converted to ethanol, acetate, and lactate, and fructose was converted to mannitol. Regarding ethanol, the concentration was relatively high in KJ-EFEL6800 and KJ-MG7011 but low in KJ-EFEL6901 and KJ-EFEL2061, revealing the superior ethanol productivity of *L. fermentum* strains.

In terms of organic acids, acetate, lactate, butyrate, propionate, and succinate were detected as the main metabolites, and their concentrations increased after fermentation. Among them, acetate was produced in the highest amount, followed by lactate, with their highest levels observed in KJ-EFEL6800 and KJ-EFEL6901. Succinate, propionate, and butyrate were also synthesized in all samples, while citrate decreased. Regarding amino acids, aspartate, alanine, arginine, glutamate, serine, and threonine were identified as major amino acids. Most amino acids showed a decrease after fermentation, except for cysteine and glycine, which increased in all kale juice samples. Particularly, the content of glutamate, which contributes to the umami taste when combined with sodium salt, significantly increased after fermentation in KJ-EFEL6901. Overall, these results indicate that, during fermentation, free sugars and amino acids in kale juice were converted to various metabolites such as alcohols (mannitol and ethanol) and organic acids (acetate, lactate, succinate, propionate, and butyrate), affecting the organoleptic and health-related properties of fermented products.

#### 3.1.5. Enzymes and Genes

The genetic analysis of EFEL6901 and EFEL6800 unveiled the presence of several enzymes, including glycoside hydrolase and glucosidase (Table 5), suggesting their potential involvement in the metabolism of kale’s abundant glycosides. The breakdown of glycosidic bonds by these enzymes would release sugars, which could serve as a carbon source in kale for the growth and proliferation of the microorganisms during fermentation. Furthermore, the increased levels of quercetin and kaempferol observed after 72 h fermentation suggest that the enzymatic activities of EFEL6901 and EFEL6800 played a significant role in enhancing the bioavailability of these bioactive compounds. This enhanced bioavailability is of particular interest, as quercetin and kaempferol are known for their potential health benefits and antioxidant properties.

### 3.2. Health-Related Characteristics of Fermented Kale Juice

#### 3.2.1. Cytotoxicity 

To assess the cytotoxicity of fermented kale juice, an MTT assay was conducted using RAW 264.7 cells. Figure 4 shows the cytotoxicity results of the tested samples. KJ-EFEL6901 and KJ-EFEL6800 showed no apparent cytotoxicity (>95% cell viability) across all concentrations tested. Meanwhile, KJ-MG7011 and KJ-EFEL2061 exhibited toxicity within 6.25–25 μg/mL and 6.25–12.5 μg/mL, respectively. This result indicates that KJ-EFEL6901 and KJ-EFEL6800 are safe with no cytotoxicity to RAW 264.7 cells within the range of 6.25–100 μg/mL.

#### 3.2.2. Antioxidant Activities 

To evaluate the antioxidative activity of fermented kale juice, DPPH and ABTS+ scavenging capacities were measured (Figure 5). In the experiment using DPPH (Figure 3A), four kale juices, KJ-EFEL6901, KJ-EFEL6800, KJ-MG7011, and KJ-EFEL2061, exhibited significantly increased activities (57.9, 58.6, 55.6, and 55.1%, respectively) compared with kale juice (26.9%) or pasteurized kale juice (29.9%). Their levels were comparable to the DPPH scavenging activity of 0.5 mM ascorbic acid (62.9%). Similarly, in the experiment using ABTS (Figure 3B), four kale juice samples showed significant increases in ABTS+ scavenging activity compared with the pre-fermentation samples. Their ABTS+ scavenging activities were in the order of KJ-EFEL6800, KJ-EFEL6901, KJ-MG7011, and KJ-EFEL2061. Their levels were also superior to comparable to the ABTS+ scavenging activity of 0.5 mM ascorbic acid. Overall, the fermentation of kale juice with the selected LAB strains resulted in a significant increase in antioxidant activity, particularly in KJ-EFEL6901 and KJ-EFEL6800. 

#### 3.2.3. NO Production

To assess the inhibitory activity of fermented kale juice on NO production in LPS-induced RAW 264.7 cells, the levels of NO in the cellular supernatants were measured (Figure 6). Treatment with LPS (1 μg/mL) significantly increased NO production compared with the control group (*p* < 0.001), while treatment with methyl arginine resulted in a dose-dependent decrease in NO production. When it comes to the fermented kale juices, their treatment significantly decreased NO production and exhibited higher inhibitory activity compared with the control juice. Among the fermented kale juices, KJ-EFEL6901 showed the highest inhibitory activity in reducing NO production (*p* < 0.01) compared with the other juices. It is noteworthy that KJ-MG7011 and KJ-EFEL2061 did not induce NO production at high concentrations (50–100 μg/mL) owing to their high cytotoxicity on RAW 264.7 cells.

#### 3.2.4. mRNA Levels of iNOS and COX-2

To analyze the anti-inflammatory effect of fermented kale juice, the transcription levels of iNOS and COX-2 in RAW 264.7 cells, which are related to the synthesis of NO or prostaglandin, were analyzed using RT-qPCR (Figure 7). The results demonstrated that treatment with LPS (1 μg/mL) significantly increased the mRNA levels of iNOS and COX-2 compared with the control group. When it comes to the fermented kale juices, their treatment significantly reduced the mRNA levels of iNOS and COX-2 in a dose-dependent manner compared with PKJ. Among the tested juices, KJ-EFEL6901 and KJ-EFEL6800 exhibited superior inhibitory activities on the mRNA transcription of iNOS and COX-2 at 6.25 and 12.5 μg/mL concentrations.

#### 3.2.5. mRNA Levels of Cytokines

To assess the anti-inflammatory effects of fermented kale juices, the transcription levels of IL-6 and IL-1β as pro-inflammatory cytokines and IL-10 as an anti-inflammatory cytokine were measured in LPS-treated RAW 264.7 cells (Figure 7). The results demonstrated that treatment with LPS (1 μg/mL) significantly increased the mRNA levels of IL-6 and IL-1β compared with the control group. In the case of fermented kale juices, their treatment significantly reduced the mRNA levels of IL-6 and IL-1β in a dose-dependent manner, compared with PKJ. At the same time, their treatment except KJ-EFEL2061 significantly increased the transcription levels of IL-10 in a dose-dependent manner compared with PKJ, revealing their superior anti-inflammatory potential. Overall, these results indicate that the kale juices fermented with selected strains exhibited anti-inflammatory activities not only by down-regulating the mRNA levels of IL-6 and IL-1β but also by up-regulating the mRNA levels of IL-10. Notably, KJ-EFEL6901 and KJ-EFEL6800 exhibited superior anti-inflammatory activities compared with PKJ, indicating the synergistic effects of kale juice and LAB starter. These findings highlight the potential of fermented kale juice to modulate the expression levels of pro-inflammatory and anti-inflammatory cytokines, suggesting its anti-inflammatory properties.

## 4. Discussion

Fermented foods are good carriers for delivering probiotics while having health-promoting effects [33]. In this study, EFEL6901 and EFEL6800 strains were selected as starters for kale juice fermentation. The two strains exhibited substantial cell growth in kale juice, reaching cell counts of >9.0 log CFU/mL (a 200-fold increment) and a pH of 4.5~4.7 after 72 h of fermentation (Figure 1). Comparing these values with the growth of *Streptococcus thermophilus*, a commonly used starter cultures for milk yogurt, which reached a cell count of 8.78 log CFU/mL and a pH of 4.3 [34], the fermentation characteristics of EFEL6901 and EFEL6800 in kale juice were comparable to the commercial product. In addition, this figure is similar to commercial kimchi (>9 Log CFU/mL), a fermented vegetable food renowned for its abundance of lactic acid bacteria [35]. The suitability of kale juice as a culture medium for selected LABs may be attributed to its rich nutrient composition. The pasteurized kale juice before fermentation contained various carbon sources such as glucose, fructose, mannose, sucrose, acetate, and citrate, and different nitrogen sources of 18 amino acids with large amounts of aspartate and glutamate (Table 3). This result indicates that kale juice can serve not only as a suitable medium for the cell growth of the selected LAB strains but also as an excellent carrier for delivering probiotics. 

Kale has abundant flavonoid content, specifically quercetin and kaempferol [29]. In this study, the initial concentrations of quercetin and kaempferol in kale juice were 7.87 μM (45.03 mg/100 g) and 54.46 μM (311.78 mg/100 g), respectively. Those levels are higher than those of collard greens (12.4 and 43.3 mg/100 g, respectively), mustard greens (8.8 and 38.3 mg/100 g, respectively), sweet potato greens (27.9 and 5.0 mg/100 g, respectively), okra (11.1 mg/100 g quercetin and non-detectable kaempferol), and green onion (non-detectable quercetin and 4.8 mg/100 g kaempferol) [9]. Furthermore, this study shows that, after fermentation of kale juice with KJ-EFEL6901 and KJ-EFEL6800, the quercetin contents were increased by three- and five-fold, respectively, and the kaempferol contents were increased by two-folds in both samples (Figure 2). Consequently, the higher level of flavonol contents in the kale juice was remarkably enhanced to superior levels by a simple fermentation process using selected LAB.

The phenolic compounds found in plants can be divided into two categories: free phenolic compounds, which are located in the vacuoles of plant cells, and bound phenolic compounds that are covalently linked to structural components of the cell wall [36]. Fermentation has emerged as a favorable method for obtaining phenolic extracts of high quality and activity from different plant sources, utilizing techniques that are both economically and environmentally friendly [37]. Changes in phenolic compounds by the fermentation process are associated with several enzymatic reactions: the hydrolysis of ester bonds that link phenolic compounds to the cell wall matrix with esterase [38], the oxidative degradation of lignin with laccase or peroxidase [39], or the deglycosylation of flavonoids to produce aglycone with β-glucosidase [40]. Several enzymes produced by microorganisms can hydrolyze the glycosidic bonds in alkyl and aryl-β-D-glucoside, and β-glucosidase is a representative enzyme that produces aglycone [41]. For example, in soybean fermentation, *L. plantarum* WCFS1 could convert soybean isoflavones from glucosides to aglycones by β-glucosidase [42]. Furthermore, *L. plantarum* CECT 748T had the ability to convert arylglucosides into their bioactive aglycone, increasing the antioxidant activity of plant foods during fermentation [43]. In the context of food, fermented onions by *L. mesenteroides* led to an increase in flavonoid aglycone content (from 0 mg/100 g to 8 mg/100 g) [44]. In addition, the fermentation of pomegranate juice by *Lacticaseibacillus casei* resulted in an increased level of quercetin-3-glucoside (from 2.7 g/L to 7.0 g/L) [45].

Likewise, the genetic analysis of EFEL6901 and EFEL6800 revealed the presence of multiple enzymes, such as glycoside hydrolase or glucosidase (Table 5), indicating their potential involvement in the utilization of kale’s abundant glycosides. These enzymes likely played a significant role in facilitating the growth of microorganisms by releasing sugars and potentially contributed to the increased levels of quercetin and kaempferol after fermentation. These findings suggest that the efficiency of bio-conversion can be influenced by the strain’s adaptability to kale juice and its ability to produce hydrolytic enzymes [46].

Antioxidants are major contributors to the functionality of the vegetable product [47]. Overall, antioxidant activity was significantly increased in fermented kale juice, and this result might be associated with the increased flavonol contents after fermentation (Figure 3). Notably, flavonol aglycones, quercetin, and kaempferol in plants are potent antioxidants that serve to protect the plant from reactive oxygen species (ROS) [48].

The anti-inflammatory effects of quercetin and kaempferol have been demonstrated through the inhibition of PGE2, COX-2, and mPGES-1 expression, which are upregulated during LPS-induced inflammation [19]. Additionally, probiotic strains have been shown to reduce the expression of pro-inflammatory cytokines, primarily TLR-mediated mechanisms, thus exerting their anti-inflammatory effects [22]. In this study, KJ-EFEL6901 and KJ-EFEL6800 resulted in a strong anti-inflammatory effect on LPS-induced RAW 264.7 cells by inhibiting NO production via down-regulating pro-inflammatory cytokines and up-regulating anti-inflammatory cytokines (Figure 4 and Figure 5). LPS activation in the mouse macrophage cell line RAW 264.7 induces an inflammatory response, producing inflammatory cytokines such as reactive oxygen species, IL-6, and TNF-α [49]. IL-6 and IL-1β are pro-inflammatory cytokines, and IL-1β induces several genes involved with inflammation and tissue destruction, such as Th1 cell responses [50]. Furthermore, IL-6 acts as a chronic stressor that can accelerate the risk of age-related diseases by promoting the differentiation and inflammation of immune cells such as T and B cells [51]. Meanwhile, IL-10 plays an anti-inflammatory role in returning the immune system to a dormant state by inhibiting activated macrophages. Notably, the results of this study suggest that the anti-inflammatory effect of LBRE6901-KJ and LBF6800-KJ was improved compared with that of kale juice, possibly due to the LAB themselves and/or their metabolites synthesized during fermentation. In our previous studies, strong antioxidative and anti-inflammatory activities of *L. reuteri* EFEL6901 and *L. fermentum* EFEL6800 have been demonstrated in in vitro and in vivo experiments [24,25]. Therefore, the superior levels of antioxidant and anti-inflammatory activities observed in KJ-EFEL6901 and KJ-EFEL6800 were attributed to the dual sources that are kale, a flavonoid-rich vegetable, and *L. reuteri* EFEL6901 and *L. fermentum* EFEL6800, probiotic starters. 

## 5. Conclusions

In this study, EFEL6901 and EFEL6800 strains exhibited superior microbial growth (>9.0 log CFU/mL) and acidity change (pH4.5~4.7) in kale juice, indicating their suitability as microbial starters for kale fermentation. After fermentation, the quercetin and kaempferol contents rapidly increased. The superior levels of antioxidant and anti-inflammatory activities observed in KJ-EFEL6901 and KJ-EFEL6800 were attributed to both kale and LAB strains in a synergistic manner. This study indicates that the kale juices fermented with EFEL6901 and EFEL6800 strains can be a superfood, serving abundant nutrients like quercetin and kaempferol as well as carrying health-promoting probiotics.

## Figures and Tables

**Figure 1 antioxidants-12-01850-f001:**
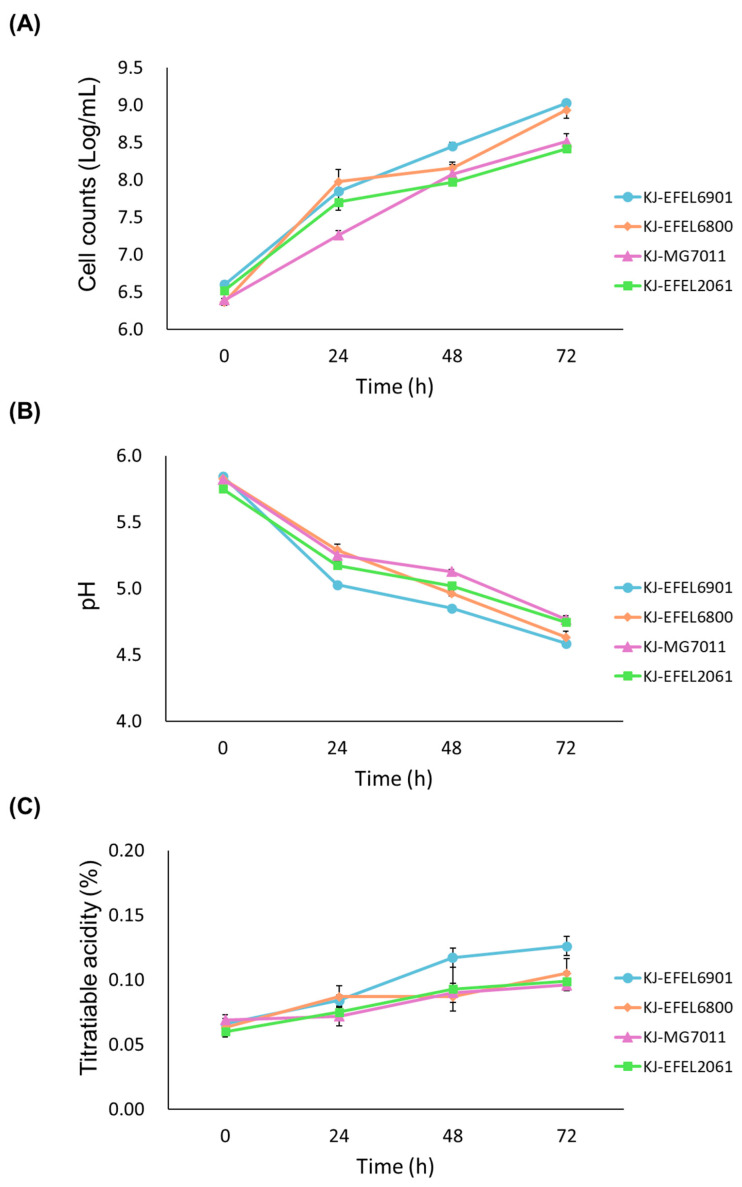
Cell growth (**A**), pH changes (**B**), titratable acidity (**C**) of kale juices fermented at 37 °C for 72 h. KJ-EFEL6901, KJ-EFEL6800, KJ-MG7011, and KJ-EFEL2061 corresponded to the kale juices fermented with *Limosilactobacillus reuteri* EFEL6901, *L. fermentum* EFEL6800, *L. fermentum* MG7011, and *Leuconostoc citreum* EFEL2061, respectively. Results are expressed as means ± standard deviations (*n* = 3).

**Figure 2 antioxidants-12-01850-f002:**
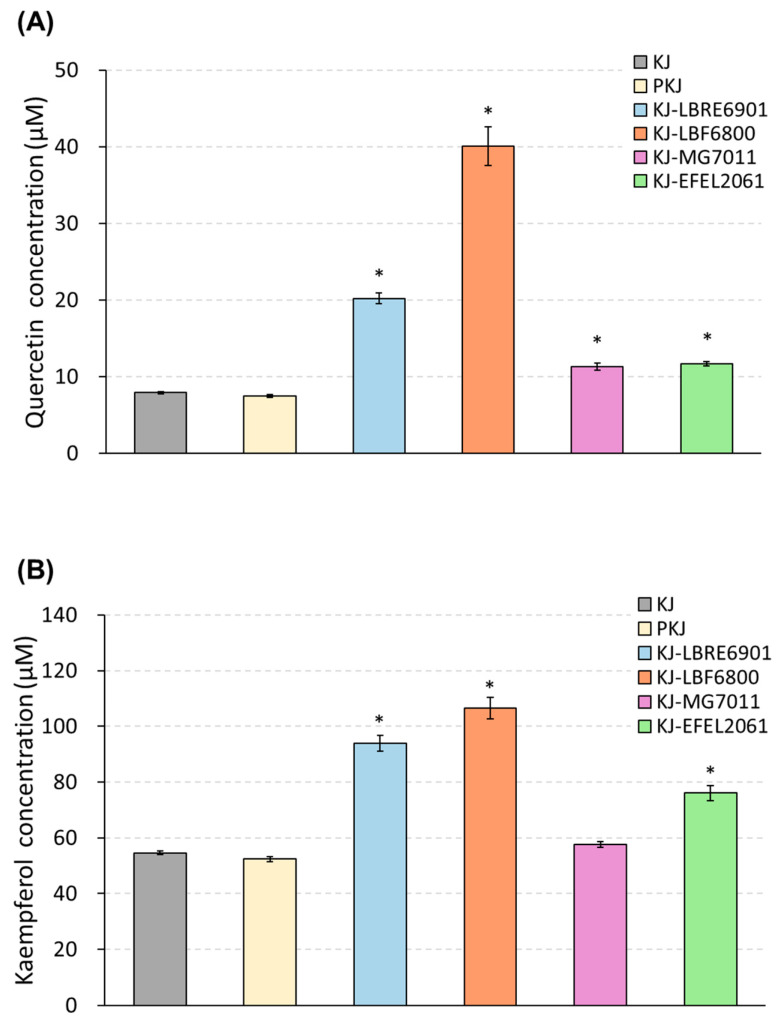
Quantification of quercetin (**A**) and kaempferol (**B**) in kale juices fermented at 37 °C. KJ, kale juice; PKJ, pasteurized kale juice before fermentation. KJ-EFEL6901, KJ-EFEL6800, KJ-MG7011, and KJ-EFEL2061 corresponded to the kale juices fermented with *Limosilactobacillus reuteri* EFEL6901, *L. fermentum* EFEL6800, *L. fermentum* MG7011, and *Leuconostoc citreum* EFEL2061, respectively. Quercetin and kaempferol contents were measured by HPLC analysis. Results are expressed as means ± SD (*n* = 3). The statistical analysis was performed using the independent *t*-test compared with PKJ (* *p* < 0.001).

**Figure 3 antioxidants-12-01850-f003:**
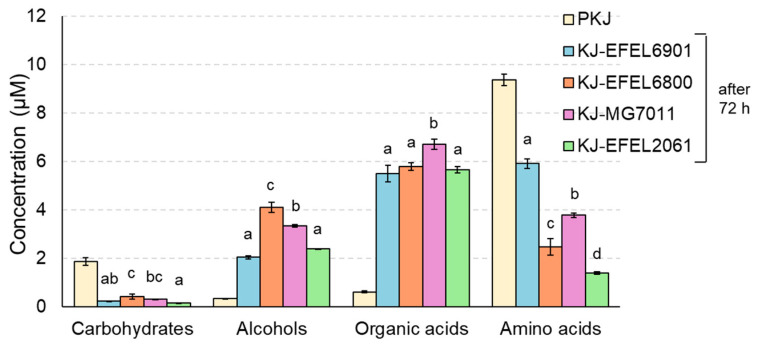
Changes in metabolite concentrations of kale juices fermented at 37 °C for 72 h. Metabolites were analyzed by ^1^H-NMR. PKJ, pasteurized kale juice before fermentation. KJ-EFEL6901, KJ-EFEL6800, KJ-MG7011, and KJ-EFEL2061 corresponded to the kale juices fermented with *Limosilactobacillus reuteri* EFEL6901, *L. fermentum* EFEL6800, *L. fermentum* MG7011, and *Leuconostoc citreum* EFEL2061, respectively. Results are expressed as means ± standard deviations (*n* = 3). Different letters indicate a significant difference in the same group at *p* < 0.05 according to Tukey’s multiple range test.

**Figure 4 antioxidants-12-01850-f004:**
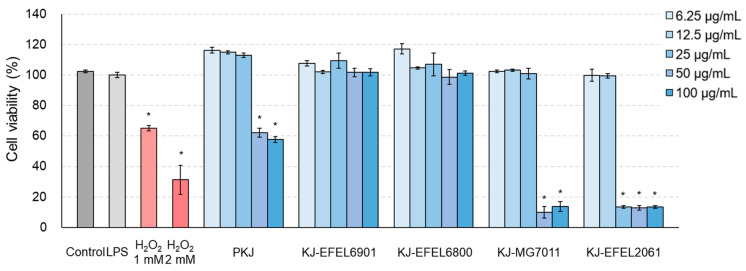
Effects of fermented kale juice on cell viability of RAW 264.7 cells. RAW 264.7 cells (2 × 10^5^ cells/well) were treated with 6.25, 12.5, 25, 50, and 100 μg/mL of fermented kale juice powder with 1 μg/mL LPS for 24 h, and cell viability was determined by the MTT assay. Dimethylsulfoxide was used as a vehicle. H_2_O_2_ was used as a negative control. LPS, lipopolysaccharide; PKJ, pasteurized kale juice before fermentation. KJ-EFEL6901, KJ-EFEL6800, KJ-MG7011, and KJ-EFEL2061 corresponded to the kale juices fermented at 37 °C for 72 h with *Limosilactobacillus reuteri* EFEL6901, *L. fermentum* EFEL6800, *L. fermentum* MG7011, and *Leuconostoc citreum* EFEL2061, respectively. Data are the mean ± SD (*n* = 3). The statistical analysis was performed using the independent *t*-test compared with LPS (* *p* < 0.05).

**Figure 5 antioxidants-12-01850-f005:**
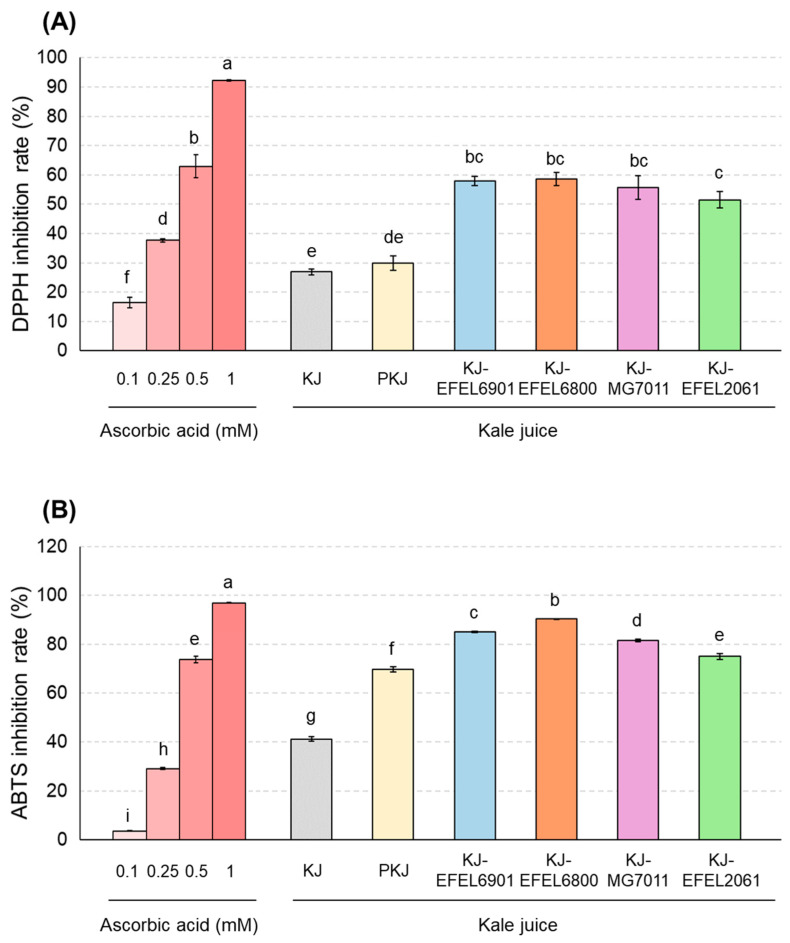
DPPH free radical scavenging activity (**A**) and ABTS+ free radical scavenging activity (**B**) of fermented kale juices. KJ, kale juice; PKJ, pasteurized kale juice before fermentation. KJ-EFEL6901, KJ-EFEL6800, KJ-MG7011, and KJ-EFEL2061 corresponded to the kale juices fermented at 37 °C for 72 h with *Limosilactobacillus reuteri* EFEL6901, *L. fermentum* EFEL6800, *L. fermentum* MG7011, and *Leuconostoc citreum* EFEL2061, respectively. Different letters indicate a significant difference at *p* < 0.05 according to Tukey’s multiple range test.

**Figure 6 antioxidants-12-01850-f006:**
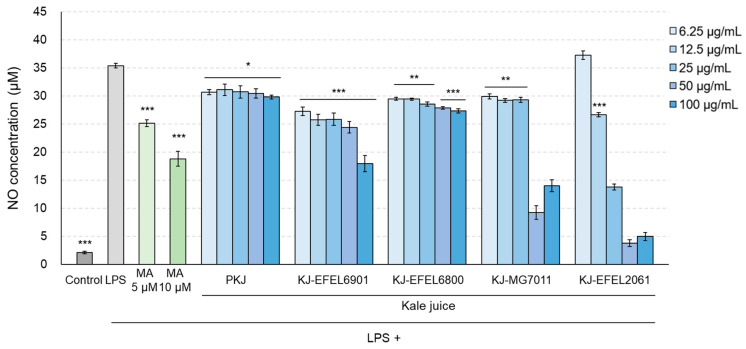
Effects of fermented kale juice on the production of NO in LPS-induced RAW 264.7 cells. KJ-EFEL6901, KJ-EFEL6800, KJ-MG7011, and KJ-EFEL2061 corresponded to the kale juices fermented at 37 °C for 72 h with *Limosilactobacillus reuteri* EFEL6901, *L. fermentum* EFEL6800, *L. fermentum* MG7011, and *Leuconostoc citreum* EFEL2061, respectively. RAW 264.7 cells were incubated with 6.25, 12.5, 25, 50, and 100 μg/mL of fermented kale juice powder with 1 μg/mL LPS for 24 h. Dimethylsulfoxide was used as a vehicle. NO was measured according to the Griess reaction. LPS, lipopolysaccharide; MA, methyl arginine (positive control); PKJ, pasteurized kale juice before fermentation. Data are the mean ± SD (*n* = 3). The statistical analysis was performed using the independent *t*-test compared with LPS (** p* < 0.05, *** p* < 0.01, **** p* < 0.001).

**Figure 7 antioxidants-12-01850-f007:**
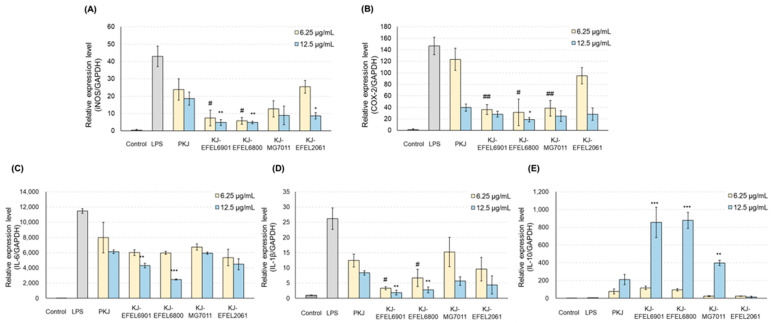
Effects of fermented kale juice on mRNA expression of iNOS (**A**), COX-2 (**B**), IL-6 (**C**), IL-1β (**D**), and IL-10 (**E**) in LPS-induced RAW 264.7 cells. RAW 264.7 cells were incubated with 6.25 and 12.5 μg/mL of fermented kale juice powder with 1 μg/mL LPS for 24 h. Dimethylsulfoxide was used as a vehicle. The mRNA expression levels of iNOS and COX-2 were determined by real-time PCR. LPS, lipopolysaccharide; PKJ, pasteurized kale juice before fermentation. KJ-EFEL6901, KJ-EFEL6800, KJ-MG7011, and KJ-EFEL2061 corresponded to the kale juices fermented at 37 °C for 72 h with *Limosilactobacillus reuteri* EFEL6901, *L. fermentum* EFEL6800, *L. fermentum* MG7011, and *Leuconostoc citreum* EFEL2061, respectively. Data are the mean ± SD (*n* = 3). The statistical analysis was performed using the independent *t*-test compared with 6 μg/mL of PKJ *(*# *p* < 0.05, ## *p* < 0.01) or 12.5 μg/mL of PKJ (* *p* < 0.05, ** *p* < 0.01, *** *p <* 0.001).

**Table 1 antioxidants-12-01850-t001:** List of strains used in this study.

Strain	Collection	Abbreviation	Culture Condition
*Limosilactobacillus reuteri* EFEL6901	KACC 81105BP	EFEL6901	37 °C, MRS
*L. fermentum* EFEL6800	KACC 81106BP	EFEL6800	37 °C, MRS
*L. fermentum* MG7011	KACC 81147BP	MG7011	37 °C, MRS
*Leuconostoc citreum* EFEL2061	KACC 92070P	EFEL2601	37 °C, MRS

**Table 2 antioxidants-12-01850-t002:** Primer sequences for mouse-specific PCR.

Gene	Forward Primer (5′-3′)	Reverse Primer (5′-3′)	Gene Accession Code
GAPDH	TTGTCTCCTGCGACTTCAACA	GCTGTAGCCGTATTCATTGTCATA	XM_059184619.1
iNOS	ACCATGGAGCATCCCAAGTA	CCATGTACCAACCATTGAAGG	NM_010927.4
COX-2	AGCATTCATTCCTCTACATAAGC	GTAACAACACTCACATATTCATACAT	AH002100.2
IL-6	GAGGATACCACTCCCAACAG	AAGTGCATCATCGTTGTTCATA	BC132458.1
IL-1β	GTTGACGGACCCCAAAAGAT	CACACACCAGCAGGTTATCA	M15131.1
IL-10	GGACAACATACTGCTAACCGACTC	AAAATCACTCTTCACCTGCTCCAC	XM_036162094.1

**Table 3 antioxidants-12-01850-t003:** Colorimetric properties of fermented kale juice.

Sample	L*	a*	b*	ΔE*
Kale juice	29.95 ± 0.02 ^c^	−6.56 ± 0.02 ^e^	10.91 ± 0.02 ^a^	6.30 ± 0.04 ^a^
PKJ	34.61 ± 0.00 ^a^	−6.80 ± 0.01 ^e^	15.14 ± 0.03 ^b^	-
KJ-EFEL6901	34.02 ± 0.01 ^b^	−0.01 ± 0.01 ^a^	13.73 ± 0.02 ^b^	6.80 ± 0.02 ^c^
KJ-EFEL6800	34.59 ± 0.03 ^a^	−1.36 ± 0.01 ^d^	15.25 ± 0.09 ^b^	5.67 ± 0.04 ^c^
KJ-MG7011	34.61 ± 0.04 ^a^	−0.76 ± 0.04 ^b^	15.70 ± 1.44 ^b^	6.49 ± 0.47 ^b^
KJ-EFEL2061	34.64 ± 0.02 ^a^	−0.85 ± 0.01 ^c^	14.65 ± 0.01 ^b^	6.07 ± 0.04 ^c^

L*—lightness, a*—redness, b*—yellowness, ΔE—color difference. Kale juice was fermented at 37 °C for 72 h. PKJ, pasteurized kale juice before fermentation. KJ-EFEL6901, KJ-EFEL6800, KJ-MG7011, and KJ-EFEL2061 corresponded to the kale juice fermented with *Limosilactobacillus reuteri* EFEL6901, *L. fermentum* EFEL6800, *L. fermentum* MG7011, and *Leuconostoc citreum* EFEL2061, respectively. The measurements were made in triplicate and are represented as mean values (±SD). Different letters indicate a significant difference at *p* < 0.05 according to Tukey’s multiple range test for each column. ΔE* was calculated based on a comparison with the L*, a*, and b* values of PKJ.

**Table 4 antioxidants-12-01850-t004:** Metabolite concentrations in fermented kale juice analyzed by ^1^H-NMR.

Group	Metabolites	PKJ	KJ-EFEL6901	KJ-EFEL6800	KJ-MG7011	KJ-EFEL2061
Carbohydrates	Fructose	0.65 ± 0.12 ^a^	0.08 ± 0.01 ^b^	0.20 ± 0.03 ^b^	0.14 ± 0.00 ^b^	0.04 ± 0.00 ^b^
	Glucose	0.90 ± 0.04 ^a^	0.06 ± 0.01 ^b^	0.06 ± 0.01 ^b^	0.05 ± 0.00 ^b^	0.04 ± 0.00 ^b^
	Mannose	0.16 ± 0.05 ^a^	0.09 ± 0.01 ^b^	0.13 ± 0.04 ^b^	0.09 ± 0.00 ^b^	0.07 ± 0.00 ^b^
	Sucrose	0.14 ± 0.01 ^a^	0.02 ± 0.00 ^b^	0.05 ± 0.04 ^b^	0.05 ± 0.00 ^b^	0.02 ± 0.00 ^b^
Alcohols	Mannitol	0.10 ± 0.01 ^ab^	1.23 ± 0.03 ^ab^	1.14 ± 0.05 ^a^	1.10 ± 0.02 ^a^	0.98 ± 0.02 ^ab^
	Ethanol	0.24 ± 0.01 ^c^	1.74 ± 0.06 ^b^	2.98 ± 0.24 ^a^	2.25 ± 0.05 ^b^	1.42 ± 0.01 ^ab^
Organic acids	Acetate	0.19 ± 0.01 ^d^	3.24 ± 0.33 ^ab^	3.54 ± 0.11 ^a^	2.63 ± 0.13 ^c^	1.76 ± 0.25 ^b^
	Succinate	0.09 ± 0.01 ^c^	0.85 ± 0.04 ^b^	0.30 ± 0.01 ^c^	1.30 ± 0.17 ^a^	1.13 ± 0.03 ^a^
	Lactate	0.09 ± 0.01 ^d^	1.69 ± 0.21 ^a^	1.86 ± 0.03 ^a^	0.97 ± 0.01 ^c^	0.49 ± 0.02 ^b^
	Propionate	0.03 ± 0.01 ^c^	0.54 ± 0.00 ^b^	0.71 ± 0.06 ^b^	0.63 ± 0.04 ^b^	0.84 ± 0.03 ^a^
	Pyruvate	0.07 ± 0.01 ^b^	0.08 ± 0.04 ^a^	0.07 ± 0.01 ^a^	0.06 ± 0.01 ^a^	0.04 ± 0.00 ^ab^
	Butyrate	0.04 ± 0.00 ^c^	0.30 ± 0.01 ^b^	0.18 ± 0.01 ^a^	0.23 ± 0.40 ^a^	0.19 ± 0.01 ^a^
	Citrate	0.12 ± 0.02 ^a^	0.02 ± 0.01 ^b^	0.04 ± 0.01 ^b^	0.02 ± 0.00 ^b^	0.01 ± 0.00 ^b^
Amino acids	Aspartate	1.40 ± 0.04 ^b^	1.08 ± 0.08 ^a^	0.04 ± 0.01 ^c^	0.03 ± 0.01 ^c^	0.06 ± 0.01 ^c^
	Glutamate	1.13 ± 0.11 ^b^	1.35 ± 0.07 ^a^	0.33 ± 0.02 ^c^	0.27 ± 0.01 ^c^	0.06 ± 0.00 ^d^
	Cysteine	0.06 ± 0.01 ^b^	0.17 ± 0.01 ^a^	0.12 ± 0.03 ^b^	0.12 ± 0.00 ^b^	0.07 ± 0.00 ^bc^
	Glycine	0.08 ± 0.03 ^bc^	0.13 ± 0.01 ^ab^	0.16 ± 0.02 ^a^	0.18 ± 0.02 ^a^	0.06 ± 0.00 ^c^
	Histidine	0.01 ± 0.00 ^a^	0.01 ± 0.01 ^a^	0.01 ± 0.00 ^a^	0.00 ± 0.00 ^a^	0.01 ± 0.00 ^a^
	Alanine	0.91 ± 0.03 ^a^	0.58 ± 0.01 ^b^	0.62 ± 0.13 ^b^	0.48 ± 0.03 ^b^	0.09 ± 0.01 ^c^
	Serine	1.18 ± 0.21 ^a^	0.09 ± 0.01 ^b^	0.12 ± 0.01 ^b^	0.12 ± 0.01 ^b^	0.05 ± 0.01 ^b^
	Threonine	0.30 ± 0.03 ^a^	0.02 ± 0.00 ^c^	0.08 ± 0.04 ^c^	0.18 ± 0.03 ^b^	0.03 ± 0.00 ^c^
	Arginine	0.78 ± 0.04 ^a^	0.05 ± 0.01 ^b^	0.03 ± 0.01 ^b^	0.02 ± 0.00 ^b^	0.07 ± 0.00 ^b^
	Proline	0.44 ± 0.15 ^a^	0.37 ± 0.10 ^a^	0.26 ± 0.03 ^a^	0.30 ± 0.04 ^a^	0.17 ± 0.00 ^a^
	Tyrosine	0.19 ± 0.01 ^a^	0.15 ± 0.00 ^b^	0.03 ± 0.00 ^c^	0.19 ± 0.01 ^a^	0.06 ± 0.00 ^d^
	Valine	0.53 ± 0.21 ^abc^	0.47 ± 0.08 ^bc^	0.07 ± 0.01 ^a^	0.40 ± 0.01 ^c^	0.20 ± 0.01 ^ab^
	Methionine	0.08 ± 0.01 ^a^	0.06 ± 0.01 ^a^	0.02 ± 0.01 ^b^	0.01 ± 0.00 ^b^	0.02 ± 0.00 ^b^
	Isoleucine	0.43 ± 0.05 ^a^	0.29 ± 0.02 ^b^	0.05 ± 0.04 ^c^	0.11 ± 0.01 ^c^	0.05 ± 0.01 ^c^
	Leucine	0.48 ± 0.02 ^a^	0.31 ± 0.07 ^b^	0.14 ± 0.00 ^c^	0.16 ± 0.00 ^c^	0.10 ± 0.01 ^c^
	Phenylalanine	0.56 ± 0.05 ^a^	0.16 ± 0.01 ^c^	0.05 ± 0.03 ^cd^	0.38 ± 0.01 ^b^	0.10 ± 0.01 ^c^
	Tryptophan	0.23 ± 0.02 ^a^	0.21 ± 0.01 ^a^	0.07 ± 0.00 ^b^	0.14 ± 0.01 ^c^	0.07 ± 0.00 ^b^
	Lysine	0.63 ± 0.03 ^b^	0.43 ± 0.05 ^c^	0.30 ± 0.02 ^c^	0.70 ± 0.01 ^b^	0.09 ± 0.00 ^a^

PKJ, pasteurized kale juice before fermentation. KJ-EFEL6901, KJ-EFEL6800, KJ-MG7011, and KJ-EFEL2061 corresponded to the kale juice fermented with *Limosilactobacillus reuteri* EFEL6901, *L. fermentum* EFEL6800, *L. fermentum* MG7011, and *Leuconostoc citreum* EFEL2061, respectively, at 37 °C for 72 h. Results are expressed as means ± standard deviations (*n =* 3). Different letters indicate a significant difference at *p* < 0.05 according to Tukey’s multiple range test.

**Table 5 antioxidants-12-01850-t005:** Genetic analysis results of EFEL6901 and EFEL6800 revealing the presence of glycoside hydrolase or glucosidase enzymes.

Related Enzyme	Gene Annotation	Predicted Mechanism	Protein ID	
			EFEL6901	EFEL6800
Glycoside hydrolase	Glycoside hydrolase family 13 protein	hydrolysis of α-glucoside linkages	WP_003668994.1	WP_023466491.1
	Glycoside hydrolase family 65 protein	hydrolysis of α-glucosidic linkages (mainly phosphorylase)	WP_003669571.1	WP_023465690.1
	Glycoside hydrolase family 70 protein	hydrolysis of glycosidic linkages (transfer D-glucopyramnosyl units from sucrose onto acceptor molecules)	WP_229392148.1	
	Glycoside hydrolase family 73 protein	hydrolysis of β-1,4-glycosidic linkage between N-acetylglucosaminyl (NAG) and N-acetylmuramyl (NAM) moieties	WP_003674860.1 WP_003669289.1	WP_023466547.1 WP_003681382.1 WP_004562972.1
	Glycoside hydrolase family 2 TIM barrel-domain containing protein	hydrolysis of β-glycosidic linkages (broad spectrum)	WP_003667314.1	WP_282347715.1 WP_282348494.1
	Glycoside hydrolase family 3 N-terminal domain-containing protein	hydrolysis single glycosyl residues from the non-reducing ends of substrate (exo type, broad spectrum)	-	WP_075667436.1 WP_240824081.1
	Family 78 glycoside hydrolase catalytic domain	hydrolysis of α-L-rhamnosyl-linkages in L-rhamnosides	-	WP_282348481.1
Glucosidase	Alpha-glucosidase	hydrolysis of α-glucoside linkages	WP_003669620.1	WP_023466314.1
	Glycosyl hydrolase 53 family protein	β-1,4-galactanase	WP_003668597.1	-

## Data Availability

Data sharing not applicable.

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
