# Peer review of "Synergistic Antioxidant and Anti-Inflammatory Activities of Kale Juice Fermented with Limosilactobacills reuteri EFEL6901 or Limosilactobacills fermentum EFEL6800"

_antioxidants, 2023, doi:10.3390/antiox12101850_

Round 1

Reviewer 1 Report

General comments

The novelties of this work, relatively at the use of lactic acid bacteria for vegetables fermentation, is limited. There are different works that prove that fermented vegetables exert better biological activities respect to a not fermented ones. Moreover, several works deal with the use of kale for obtained new fermented products (Kim, 2017; Michalak, Kubik-Komar, Waśko, & Polak-Berecka, 2020; Szutowska & Gwiazdowska, 2021; Szutowska, et al., 2021; Szutowska, Rybicka, Pawlak-Lemańska, & Gwiazdowska, 2020).

The authors, for my advice, should report which are the new results of their work respect to knowledge already acquired.

Figures and tables is not clear presented. Furthermore, I have many doubts about how the statistical analysis of the data was done. In my opinion the authors should be used only the ANOVA for the analysis of data. In addition, it is necessary to correct for multiple comparisons using Bonferroni or by controlling the false discovery rate (FDR).

The discussion of the data should be improved by including data obtained on others fermented vegetables.

Abstract

L23 “NO” all acronyms should be reported in extenso at their first appearance in the text (please check throughout the text for all others acronyms)

L24 the authors should report the markers and cytokines studied in this work

Material and methods

L96 what do the authors mean by probiotic activities? which probiotic activities?

L110 The presence of thermoduric spore forming bacteria should be evaluated. Moreover the authors did not check the purity of strains utilized at the end of fermentation.

L120 here or in the discussion section, the authors should explain because is important to evaluate the colour changes of the fermented kale.

L154 Describe and explain how freeze-drying was done.

L169 the authors should explain how they made the calibration curve.

L183 “2.7. Cytotoxicity of fermented kale juice” Please rewrite completely this paragraph is not very clearly presented (check the experimental design, acronyms, formula …).

L214 “transcriptome” is the set of all RNA transcripts, including coding and non-coding, in an individual or a population of cells. The authors in this work evaluated the expression of some genes, therefore please check and revise this sentence.

L216-217 please check and revise this sentence.

L223 please report the quantity of cDNA used in real time PCR reaction

Table 2: In the materials and methods section in Table 2, the authors among the genes studied include TNA alpha, but no results concerning this gene are provided. While results are given for the IL 6 gene, which is not listed among the genes analysed. Please check and analyse.

L233 Indicate the genomes references of the strains used and, on which strains the genomic analysis was performed. As supplementary material you will added the sequences obtained.

L238 In the statistical analysis paragraph the authors report for each experiment the factor considered for ANOVA analysis.

Results

L293 (figure caption) in the same column ? or in the same row?

L295-296 please check this sentence, is not clear. Explain it better in the materials and methods section.

L342 L.fermentum in italic

Table 4 this table as presented is unreadable. Please see my previous comment.

L372 The authors for sake of clarity should be report the references indications of gene/genome sequences of the strains used in this work present in databanks. Indicate also the probably mechanisms of action and the substrate of each enzymes reported in the table 

L400 report this range also in material and methods section.

Discussion

L497 the authors should be compared their data, with the data obtaining to other authors on the same vegetable of on others vegetables.

Author Response

Thank you for your kind comments. We focused on synergistic effect of kale and lactic acid bacteria, especially on antioxidant and anti-inflammatory effects. During the process, we demonstrated that kale fermentation with lactic acid bacteria increased the bioactive compounds, quercetin and kaempferol, enhancing their antioxidant and anti-inflammatory effects at an in vitro Furthermore, through genetic analysis of the lactic acid bacteria, we revealed novel insights into which enzymes and mechanisms were involved in increasing the levels of quercetin and kaempferol. We improved these aspects in the Results and Discussion sections. The revised manuscript is presented in blue as below.

Reviewer 2 Report

Minor editing of English required which can be covered by the editorial board or the authors. A few aspects to consider:

Table 1, Column, title should be strains and not species.

Lines 35-36: write clearly about A sample = absorbance ............; A control = absorbance of distilled water added to sample

Equation should be labelled as Equation 1, etc to equation 2 on page on the following page.

Line 56: delete for twice. Delete And 

2.9 The results were analysed after normalisation...............

Figure 2 adding legends would help with assorted colours. 

Add: ......at 72 h in the Figure Title and in Figure 3

In the results and discussion - it is important to note that, your results are for the beginning of fermentation (time 0) and end of fermentation at 72 h. Therefore, discussion about what happens during fermentation need to be done with some caution as that is based on the science of fermentation by the strains.

3.1.5 Enzymes and genes

Line 418: Start with, similarly.................

As a rule, when using the term significantly in the Results and Discussion, the p-value should be added.; e.g. line 519

Line 543 -------by microorganisms

Conclusion:

Delete proper acidity, line 598

Delete dramatically, and replace with rapidly, line 601

High Quality

Author Response

Thank you for your comment. We improve the English quality of the manuscript accordingly. The revised manuscript is presented in blue as below.    

Round 2

Reviewer 1 Report

The authors have taken into consideration all the issues raised previously, the work could be suitable for publication in the Antioxidant journal